**Daily black carbon emissions from fires in Northern Eurasia for 2002–2015**
W. M. Hao[1*], A. Petkov[1], B. L. Nordgren[1], R. Corley[1], R. P. Silverstein[1], S. P. Urbanski[1], N.
Evangeliou[2,3], Y. Balkanski[2], and B. Kinder[4]
[1]Missoula Fire Sciences Laboratory, Rocky Mountain Research Station, United States Forest
Service, Missoula, Montana, USA
[2]Laboratoire des Sciences du Climat et de l'Environnement (LSCE), CEA-UVSQ-CNRS UMR 8212,
Institut Pierre et Simon Laplace, L'Orme des Merisiers, F-91191 Gif sur Yvette Cedex, France
[3]Norwegian Institute for Air Research (NILU), Department of Atmospheric and Climate Research
(ATMOS), Kjeller, Norway
[4]International Program, United States Forest Service, Washington, D.C., USA
[*]Corresponding author: US Forest Service, Rocky Mountain Research Station, Fire Sciences
Laboratory, 5775 Highway 10W, Missoula, Montana 59808, USA. T: +1-406-329-4838, F: +1-
406-329-4867, whao@fs.fed.us
**Abstract**
Black carbon (BC) emitted from fires in Northern Eurasia is transported and deposited on ice and
snow in the Arctic and can accelerate its melting during certain times of the year. Thus, we
developed a high spatial resolution (500 m × 500 m) dataset to examine daily BC emissions from
fires in this region for 2002–2015. Black carbon emissions were estimated based on MODIS land
cover maps and detected burned areas, the Forest Inventory Survey of the Russian Federation,
the IPCC Tier-1 Global Biomass Carbon Map for the year 2000, and vegetation specific BC
emission factors. Annual BC emissions from Northern Eurasian fires varied greatly, ranging
from 0.39 Tg in 2010 to 1.82 Tg in 2015, with an average of 0.71±0.37 Tg from 2002–2015.
During the 14-year period, BC emissions from forest fires accounted for about two-thirds of the
emissions, followed by grassland fires (18%). Russia dominated the BC emissions from forest
fires (92%) and Central and Western Asia was the major region for BC emissions from grassland
fires (54%). Overall, Russia contributed 80% of the total BC emissions from fires in Northern
Eurasia. Black carbon emissions were the highest in the years of 2003, 2008, and 2012.
Approximately 58% of the BC emissions from fires occurred in spring, 31% in summer, and
10% in fall. The high emissions in spring also coincide with the most intense period of ice and
snow melting in the Arctic.

## 1   Introduction

Black carbon (BC), a major component of light absorbing aerosols, is one of the leading species for climate forcing (IPCC 2013; Bond et al., 2013; Stohl et. al., 2015; Sand et al., 2016). Black carbon absorbs solar radiation, affects radiative forcing, and causes warming of the atmosphere. Black carbon deposited on the Arctic and mountains can accelerate the melting of snow (Flanner et al., 2007). The two most recent estimates of BC global sources agree well: 7.5 (2.0–29) Tg yr$^{-1}$ for the year 2000 (Bond et al., 2013) and 7.7 ± 1.0 Tg yr$^{-1}$ (Wang et al., 2014) for the years 2000–2007 (1 Tg = $10^{12}$ g). These estimates were also consistent with an earlier estimate of 8.0 (4.3–22) Tg yr$^{-1}$ (Bond et al., 2004). Open biomass burning accounts for about 37% of the BC sources (e.g., Bond et al., 2013; Wang et al., 2014) whereas other combustion processes (fossil fuels, transportation, industry, power generation, domestic biofuels) account for the balance. Black carbon is an ideal target for mitigation of global warming because of its short atmospheric lifetime of about a week.

Deposition of BC on Arctic ice and snow has major impacts on global climate. Black carbon deposited on ice and snow absorbs solar radiation that leads to reduced surface albedo, accelerated melting of ice and snow, and increased sea levels (Warren and Wiscombe, 1985; Clarke and Noone, 1985; McConnell et al., 2007). Biomass burning has been identified to be the dominant source of BC in the Arctic during spring (Stohl et al., 2006; Treffeisen et al., 2007; Hegg et al., 2009; Warneke et al., 2009; Hegg et al., 2010; Warneke et al., 2010; Bian et al., 2013), the most prevalent period for snow melting and Arctic Haze events (e.g. Quinn et al., 2007). The fires usually occur in the boreal forests and agricultural lands of Northern Eurasia. Black carbon emitted from boreal forest fires in North America in summer can also be deposited on Arctic snow and reduce surface albedo (Stohl et al., 2006). These findings were based on episodic events observed from airborne campaigns, ground-based monitoring, and dispersion modeling. However, they do not provide the spatial and temporal variability and the specific amount of BC emitted from various biomass burning sources (e.g., forest, grassland, shrubland, savanna, and cropland). Such information is critical for assessing the impacts of BC on accelerated melting of Arctic ice and snow and on solar radiation in the atmosphere. In addition, BC deposition on the Arctic is further complicated by the dome effects of atmospheric circulation that limits the transport of air masses from lower latitudes into the Arctic (Stohl, 2006). Only certain weather patterns allow the transport of pollutants to the Arctic. It is therefore necessary to develop daily emission sources for the assessment of the transport and deposition of BC on Arctic ice and snow.

The **G**lobal **F**ire **E**missions **D**atabase (GFED4, http://www.globalfiredata.org; Giglio et al., 2013) provided the most detailed fire emission inventory daily or monthly at a spatial resolution of 0.25° × 0.25° globally for 1997–2015. This dataset has been widely used to study the effects of fires on atmospheric chemistry, air quality, and climate. However, it underestimated the seasonality of atmospheric aerosols in the Arctic in comparison to ground-based and satellite observations (e.g. Stohl et al., 2015).

In this study, we developed the **F**ire **E**mission **I**nventory–**N**orthern **E**urasia (FEI-NE), a dataset
of daily BC emissions from forest, grassland, shrubland, and savanna fires over Northern Eurasia
at a 500 m × 500 m resolution for 2002–2015. We examined the spatial and temporal variability
of BC emissions from fires in different ecosystems in the geopolitical regions of Russia, Eastern
Asia, Central and Western Asia, and Europe. The estimates of BC emissions in different regions
will assist policy makers in developing effective mitigation policies for reducing BC emissions
from fires and reducing the BC impacts on accelerated ice and snow melting in the Arctic.
## 2    Methods
### 2.1    Emission calculation
We define Northern Eurasia to encompass Russia, Eastern Asia, Central and Western Asia, and
Europe (Fig. 1 inset map). It covers the region of 35°N–80°N and 10°W–170°E (Fig. 1).
Emissions of BC (E) at any spatial and temporal scales are calculated by the equation (Seiler and
Crutzen, 1980; Urbanski et al., 2011):
$E = A \times FL \times \alpha \times EF$
where E is the amount of emitted BC, A is the area burned, FL is the fuel loading, $\alpha$ is
combustion completeness, and EF is the emission factor for BC. Fuel consumption is calculated
as the product of fuel loading and combustion completeness (FL × $\alpha$). We will discuss the
derivation of each parameter in the following sections.
### 2.2    Burned area
Daily area burned over Northern Eurasia was mapped at a 500 m × 500 m resolution from 2002–
2015 based on three MODIS (**MOD**erate Resolution **I**maging **S**pectroradiometer) products from
the NASA Terra and Aqua satellites (Li et al., 2004; Urbanski et al., 2009). The burned area
algorithm combines the MODIS thermal anomalies product (MOD14 for Terra and MYD14 for
Aqua) at a 1 km resolution four times daily and the MODIS top of the atmosphere calibrated
reflectance product (MOD02) to map and date burn scars. The burned area mapping method,
which was originally developed for the western United States with an uncertainty of ≤5%
(Urbanski et al., 2011), has two steps. First, a burn scar algorithm is applied to pixels of the
reflectance product to identify potential burn scars. Then, the potential burn scars are screened
for false detections using a contextual filter that eliminates pixels not proximate with recent
active fire detections. For mapping burned areas in Northern Eurasia, the original burn scar
algorithm was unchanged; however, the contextual filter was modified. In this study, potential
burn scars not within 5 km and 10 days of active fire detection were classified as false detections
and were eliminated. For the western United States, the thresholds of the contextual filter were 3
km and 5 days. Land cover classification of burned areas was based on the MODIS land
cover/land cover change product (MOD12) at a 500 m resolution (Friedl et al., 2010). The date
of a burned pixel in FEI-NE was taken as the first date the pixel satisfied the contextual filter.

1. There is no comprehensive geospatial dataset of large fires ($>4$ km$^2$) in Northern Eurasia, as in
2. the United States, to compare with our MODIS-derived burned areas. The FEI-NE algorithm for
3. mapping burned areas in Northern Eurasia had to be validated by comparison with selected
4. Landsat images (Hao et al., 2012). The high resolution burned areas were produced from Landsat
5. images acquired before and after large fires over eastern Siberia in 2001 and 2003 and compared
6. with our MODIS derived burned areas in 18,754 grid cells of 3 km $\times$ 3 km. The linear
7. relationship of our MODIS-derived burned areas vs. Landsat burned areas was slope = 1.0 with
8. an $r^2$ of 0.70.

## 2.3    Fuel loading

10. Since limited information was available on the fuel loading for different land cover types over
11. Northern Eurasia, we developed a fuel loading dataset for forested and non-forested areas over
12. Northern Eurasia at a 500 m $\times$ 500 m resolution circa 2010. The data sources were: (1) the
13. MODIS land cover map (MOD12, v5), (2) a 2010 land cover map at a 250 m resolution over
14. Russian Federation provided by the Space Research Institute (SRI) of the Russian Academy of
15. Sciences (RAS), (3) a map of dominant forest species for 2010 at a 250 m resolution over
16. Russian Federation provided by the SRI, (4) the 2003 Forestry Inventory Survey of Russian
17. Federation, and (5) the IPCC Tier-1 Global Biomass Carbon Map for the year 2000. Fuel loading
18. for forest was categorized into coarse woody debris (CWD), shrub, lower layers, litter, and duff.
19. CWD included fallen logs and branches. Lower layers referred to seedlings, dwarf-shrubs, herbs,
20. mosses and lichen (Alexeyev and Birdsey, 1998). Duff layers were measured up to 20 m deep.
21. For each of the 87 oblasts of the Russian Federation, the loading of each fuel component was
22. estimated based on the 2003 Forestry Inventory Survey of the Russian Federation provided by V.
23. Alexeyev at the RAS Sukachev Institute of Forest in Krasnoyarsk, Russia. In addition, the
24. loading of each fuel component over seven fire prone regions (northern, central, and southern
25. Krasnoyarsk, Sakha, Irkutsk, Chita, and Amur) were further characterized by different ecological
26. zones, according to Alexeyev and Birdsey (1998). The fuel loading of forested areas beyond the
27. borders of the Russian Federation was extrapolated from the closest land cover types in the
28. Russian Federation.

29. The fuel loading of non-forested areas at a 1 km $\times$ 1 km resolution was derived from the IPCC
30. Tier-1 Global Biomass Carbon Map for the year 2000 (Ruesch and Gibbs, 2008). The data
31. product was based on biomass carbon stored in aboveground living vegetation created using the
32. International Panel on Climate Change (IPCC) Good Practice Guidance for reporting national
33. greenhouse gas inventories (Penman et al., 2003).

## 2.4    Combustion completeness

35. Combustion completeness was estimated using the empirical fire effects model CONSUME
36. (Prichard et al., 2006). The CONSUME natural fuel algorithms include predictive equations for
37. the consumption of multiple fuel components: dead woody debris, shrubs and herbaceous

vegetation, litter, and duff/organic soil. In addition to mass loadings for the different fuel
components, CONSUME requires the moisture content of fine woody debris (diameter < 7.6 cm;
FMFWD), coarse woody debris (diameter > 7.6 cm; FMCWD), and duff (FMDUFF) as input. In
the FEI-NE simulations, we set the fuel moisture values to levels typical of western United States
and Canadian wildfire season conditions (FMFWD = 10%, FMCWD = 15%, FMDUFF = 40%).
The average combustion completeness predicted for forest fuels using the CONSUME
algorithms was 72% for dead woody debris, 90% for herbaceous and shrub fuels, and 58% for
combined litter and duff. As a check on the assumed fuel moistures used in our consumption
calculations, the WFDEI meteorological forcing dataset (Weedon et al., 2014) was used to
estimate FMFWD and FMCWD using the National Fire Danger Rating System basic equations
(Cohen and Deeming, 1985).  We found the areas affected by fire in Russia had average values
of FMFWD = 6% and FMCWD = 12%. To gauge the sensitivity of the fuel consumption
estimates to the fuel moisture content, we conducted a set of simulations with fuel moisture set at
twice our best estimate values. The effect reduced combustion completeness to 56% for dead
woody debris and 50% for litter and duff. The amount of the fuel burned, or fuel consumption,
was estimated as the product of fuel loading and combustion completeness.
**2.5    Emission factors**
Limited information is available on the emission factors of BC from biomass burning in
Northern Eurasia. Therefore, we used emission factors for refractory BC (rBC) from aircraft
measurements of emissions from different types of fuels in the United States (May et al., 2014).
The rBC was the refractory material in the absorbing aerosol measured by the Single Particle
Soot Photometer (SP2). The emission factors for rBC used for estimation of BC emissions were
0.93 g kg$^{-1}$ and 1.36 g kg$^{-1}$ for forest and non-forest fires, respectively.
**2.6    Uncertainty**
It is difficult to estimate the uncertainty of BC emissions from fires over diverse ecosystems of
Northern Eurasia. There is a lack of (1) ground-based surveys of fire perimeters to validate
satellite derived burned areas, (2) the methodology and field data of different fuel components in
various ecosystems to estimate fuel loading, and (3) data of combustion completeness and
emission factors from field measurements in different ecosystems. Therefore, our "best"
estimates for the uncertainty of burned areas, fuel loading, combustion completeness, and
emission factors are 30%, 50%, 20%, and 15%, respectively. The overall estimate would be
32 63%.

**3    Results**
In this section, we present for 2002–2015 the comparison of the burned areas, excluding
agricultural fires, of FEI-NE with GFED4 and the NASA's official Collection 5.1 burned area
product MCD45 (http://modis-fire.umd.edu/pages/BurnedArea.php; Roy et al., 2008). The fuel
consumption was compared for different land cover types during the 14-year period. The spatial

(500 m, regional, continental) extent and temporal (daily, seasonal, interannual) variability of BC emissions from biomass burning in Northern Eurasia and the BC emissions from fires over different land cover types and geographic regions are reported.

## 3.1    Comparison of Burned areas of FEI-NE vs. GFD4 and MCD45

During the 14-year period of 2002–2015, the annual area burned in Northern Eurasia varied considerably, ranging from $1.1 \times 10^5$ km$^2$ in 2013 to $4.9 \times 10^5$ km$^2$ in 2003, with a mean of $(2.6\pm1.0) \times 10^5$ km$^2$ yr$^{-1}$ (Fig. 2). For comparison, the total areas burned in FEI-NE, GFED4, and MCD45 were 3.6, 1.9, and $2.2 \times 10^6$ km$^2$, respectively. There were linear declines of the areas burned during this period for FEI-NE ($r^2 = 0.52$, $p = 0.003$), GFED4 ($r^2 = 0.35$, $p = 0.03$), and (MCD45) ($r^2 = 0.38$, $p = 0.02$). The rates of decrease were 16.7, 7.6, and $6.8 \times 10^3$ km$^2$ yr$^{-1}$ for FEI-NE, GFED4, and MCD45, respectively.

The interannual variability in our burned area agrees well with GFED4 and MCD45 during the 14-year period of 2002–2015 (Fig. 2), but our total burned areas were 1.8 times higher than the burned areas of GFED4 and 1.7 times higher than those of MCD45. The differences are narrowing over time. Figures 3a and 3b illustrated the geographic differences in the areas burned in Russia, Eastern Asia, Central and Western Asia, and Europe with the largest difference in the year 2003 and the smallest difference in the year 2013. It is difficult to explain the differences from one year to the other. However, it is worth noting that GFED4 uses the MODIS Collection 5.1 MCD64A1 and that in the more recent MODIS Collection 6 for MCD64A1 the total burned areas over boreal Asia have been increased by 34.7% in 2006 (Giglio, 2016). It is therefore essential to compare FEI-NE burned areas with revised GFED4 and MCD45 after the Collection 6 becomes available for all the years of 2002–2015.

## 3.2    Fuel Consumption

Fuel consumption was calculated as the product of fuel loading and combustion completeness. The fuel consumption for different land cover types over Northern Eurasia from 2002–2015 is summarized in Fig. 4. The average fuel consumption for each year was the mean of the fuel consumption for all 500 m $\times$ 500 m grid cells by land cover types. Fuel consumption was highest in the forested area [$7.7\pm3.7$ kg m$^{-2}$ (n= 14)], followed by savanna [$5.3\pm4.2$ kg m$^{-2}$ (n=14)], shrubland [$3.2\pm3.6$ kg m$^{-2}$ (n=14)], and grassland [$0.6\pm1.2$ kg m$^{-2}$ (n= 14)]. Grassland and forest fires dominated the area burned in Northern Eurasia. However, the fuel consumption per unit area in the forest is about 13 times higher than the fuel consumption in grassland.

## 3.3    Spatial distribution of daily BC emissions

The spatial distribution of daily BC emissions from biomass burning over Northern Eurasia at a 500 m $\times$ 500 m resolution in 2003 is shown in Fig. 1. The year 2003 had the largest area burned during the period of 2002–2015. Black carbon emissions in Russia were prevalent along the Trans-Siberian Railway (Fig. 1). Human activities in the villages along the railway were

probably the major cause of the fires. Figure 5 shows the maps of daily BC emitted from fires in Northern Eurasia at a 500 m × 500 m resolution for 2002–2015. Most of the BC was emitted from forest fires in Russia. Much lower emissions were produced from grassland fires in Kazakhstan. Fuel consumption in non-forested areas is substantially lower than that in forested areas (see section 3.2), even though it covered large areas burned. The spatial distribution of BC emissions in the grassland areas of Kazakhstan repeated annually (Fig. 5), suggesting the grassland was burned frequently as in the African savannas.

Table 1 summarizes the BC emissions from fires in different land cover types over different geographic regions for 2002–2015. During the 14-year period, a total of 9.9 Tg of BC were emitted. Annual BC emissions from fires varied by a factor of five ranging from 0.39 Tg in 2010 to 1.82 Tg in 2003 with an average of (0.71±0.37) Tg. About two-thirds (65%) of the emissions occurred from fires in forest, followed by grassland (18%), savanna (10%), and shrubland (7%). Geographically, approximately 92% of BC emissions from forest fires originated in Russia. For BC emissions from grassland fires, 54% occurred in Central and Western Asia and 32% in Russia. Russia also dominated the BC emissions from shrubland fires (87%) and savanna fires (83%). Overall, Russia accounted for 80% of the total BC emissions from fires in Northern Eurasia, followed by Central and Western Asia (11%).

## 3.4     Interannual variability of BC emissions

There was significant interannual variability of BC emissions in Northern Eurasia during the 14-year period of 2002–2015 (Table 1 and Fig. 6). The interannual variability of BC emissions for different land cover types followed the same variable pattern as total emissions. Grassland fires were the only land cover types to have apparent trends of BC emissions from fires, decreasing linearly at a rate of 8.8 Gg yr$^{-1}$ from 2002–2015 ($r^2 = 0.6$, $p = 0.002$, $n = 14$, Fig. 6).

Annual BC emissions for the peak three years of 2003, 2008, and 2012 were 1.82, 1.16, and 0.94 Tg, which were 156%, 64%, and 33%, respectively, above the 14-year mean for BC emissions. There was a declining trend of BC emissions for the three peak years during the 14-year period. Black carbon emissions from forest fires accounted for ~69% and grassland fires ~13% of the total BC emissions for each of the three peak years.

## 3.5     Seasonality

Daily BC emissions from fires in Northern Eurasia for each year from 2002–2015 are shown in Fig. 7. The start and end dates of BC emission periods were different for each year. During the 14 years, on average about 58% of BC was emitted in spring (March, April, May), 31% in summer (June, July, August), 10% in fall (September, October, November), and 1% in winter (December, January, February). The seasonality of BC emissions from fires in different land cover types varies considerably. The majority of emissions from forest fires occurred from late March to late May (Fig. 8a), which coincides with the forest fire season in Russia. Spring is the most effective season for acceleration of ice and snow melting by BC emissions in the Arctic

(Bond et al., 2013). This period also corresponds to the late part of the Arctic Haze season when
meteorology is favorable for the transport of emissions from lower latitudes to the Arctic region
(Quinn et al., 2007). Black carbon emissions from grassland fires over Northern Eurasia have
bimodal temporal distributions from late March to late June and from late July to the end of
October (Fig. 8b).

## 4  Discussion

We present the daily spatial and temporal distribution of BC emissions from biomass burning
over Northern Eurasia at a 500 m × 500 m resolution from 2002–2015 in Fig. 5. This BC
emission inventory is essential for modeling air quality in high latitudes and ice and snow
melting in the Arctic. The dataset has been used for studying the transport and deposition of BC
on the Arctic from 2002–2013 (Evangeliou et al., 2016). The study found that approximately
8.2±2.7% of the BC emitted by Northern Eurasian fires was deposited on the Arctic ice during
the period of 2002–2013, accounting for 45%–78% of the BC deposition from all the sources
(Evangeliou et al., 2016). About 42% of the BC emitted during spring and summer was
deposited on Arctic ice, which is the most effective period for acceleration of ice and snow
melting.

## 4.1  Emission inventory

Biomass burning in Northern Eurasia is a significant component of the global BC emission
inventory. The annual mean of our BC emission inventory in this region from 2002–2015 was
(0.71±0.37) Tg yr$^{-1}$. Based on the BC emission inventories (Bond et al., 2013; Wang et. al.,
2014) (see section 1), we estimated that wildfires in Northern Eurasia contributed about 9.2%–
9.5% of the global sources of BC and about 26% of the biomass burning source worldwide.
We compared our BC emission estimates from biomass burning sources, excluding agricultural
fires, in Northern Eurasia with GFED4.1 for 2002–2015. During the 14-year period, the
interannual variability of FEI-NE and GFED4.1 are similar, but the magnitude of BC emissions
are significantly different (Fig. 9). Total FEI-NE annual BC emissions ranged from 2.3 times
higher than those of GFED4.1 in 2010 to 4.9 times in 2003, with an average of 3.2 times.
For forested areas, BC emissions estimated by FEI-NE ranged from 2.4 times higher than
GFED4.1 in 2002 to 7.4 times higher in 2004, with an average of about four times during the 14-
year period (0.46 Tg yr$^{-1}$ for FEI-NE vs. 0.10 Tg yr$^{-1}$ for GFED4.1) (Fig. 10a). The largest
relative difference in BC emissions was in non-forested (grassland, shrubland, and savanna)
areas (Fig. Fig. 10b). The mean estimates of FEI-NE and GFED4.1 were 0.25 Tg yr$^{-1}$ and 0.016
Tg yr$^{-1}$, respectively, for the 14-year period. The differences between FEI-NE and GFED4.1 can
be attributed to the area burned (section 3.1), fuel loading/consumption (section 3.2), and the
emission factors used. The emission factors used in our inventory, 0.93 g kg$^{-1}$ for forest and 1.36
g kg$^{-1}$ for non-forest, were higher than the GFED4.1 recommended emission factors of 0.52 g kg$^{-}$
[1] for boreal forest and 0.37 g kg$^{-1}$ for savannas, grassland, and shrubland, which were from Akagi
et al. (2011).
Northern Eurasia can be categorized as a northern region dominated by forest and a southern
region dominated by grassland. Black carbon emissions from fires estimated by FEI-NE and
GFED4.1 were compared in two geographic regions: (1) Russia of FEI-NE vs. Boreal Asia
(BOAS) of GFED4.1 (Fig. 11a), and (2) Eastern Asia, Central and Western Asia, and Europe of
FEI-NE vs. Central Asia (CEAS) and Europe (EURO) of GFED4.1 (Fig. 11b). These geographic
regions defined for FEI-NE and GFED4.1 largely overlap, but there are minor discrepancies. In
Fig. 11a, forest is the dominant vegetation type and the BC emissions were dominated by forest
fires. During the 14-year period of 2002–2015, an average of 569± 357 Gg yr$^{-1}$ were emitted in
the FEI-NE Russia region compared with 106±65 Gg yr$^{-1}$ emitted in the Boreal Asia of
GFED4.1. In Fig. 11b, grassland dominated, so BC emissions from fires here were much less
than in the forested area (Fig. 11a). Only 140±48 Gg yr$^{-1}$ of BC were emitted in Eastern Asia,
Central and Western Asia, and Europe of FEI-NE, while 29±6 Gg yr$^{-1}$ were emitted in Central
Asia and Europe according to GFED4.1.
**4.2     Seasonality**
Black carbon emissions in spring have the greatest impacts on the melting of ice and snow in the
Arctic (Flanner et al., 2007; Flanner et al., 2009; Hegg et al., 2010; Bond et al., 2013). This event
usually occurs in late March. High BC concentrations in spring have been observed in smoke
plumes from aircraft measurements (Warneke et al., 2009, Warneke et al., 2010) and at the
ground monitoring station Zellepin in Norway (Stohl et al., 2007). Our estimates of BC
emissions were consistent with the observations, being the highest in spring every year from
2002–2015, though the start and end dates of BC emissions from biomass burning varied (Fig.
7). Forest fires dominated the emissions in spring (Fig. 8a). The timing and the magnitude of BC
emissions depend on the burned area and fuel conditions, which are ultimately determined by
weather and human activities. The grassland fires over Northern Eurasia often occurred in two
distinct periods: late March to late June and late July to late October.
**4.3     Russia**
One of the objectives of this study was to identify the geographic regions of BC emissions from
Northern Eurasia to support the development of mitigation policies. Russia was the dominant
region for BC emissions from biomass burning during the 14-year period, accounting for 80% of
the total emissions from fires in Northern Eurasia (Table 1). In Russia, 75% of the BC emissions
occurred in forest, 10% in savannas, 8% in shrubland, and 7% in grassland.
Spring is the most critical season for accelerated melting of ice and snow in the Arctic. Spring
fires accounted for an average of 56±17% of the annual BC emissions in Russia during the 14-
year period, followed by fires in summer (33±17%).

## 4.4       Agricultural vs. non-agricultural fires

One of the key aspects for developing mitigation policies of BC impacts on accelerated ice and snow melting in the Arctic is to understand the contribution of different biomass burning sources for BC, especially non-agricultural versus agricultural fires. It is much more feasible to control agricultural fires than wildfires. Several episodic events indicated that BC emitted from agricultural fires may be transported to the Arctic. The exceedingly high levels of equivalent BC observed at the Zellepin monitoring station in Norway in early May 2006 were due to the transport of smoke plumes from agricultural fires in Eastern Europe to the European Arctic (Stohl et al., 2007). Smoke plumes from agricultural burning in Kazakhstan and southern Russia in April 2008 have been observed to reach to the western Arctic (Warneke et al., 2009, Warneke et al., 2010; Bian et al., 2013).

The most comprehensive study of BC emissions from agricultural burning in Russia covers the period of 2003 –2009 (McCarty et al., 2012).  The annual emissions ranged from 0.002 to 0.022 Tg with an average of 0.009 Tg, in which about 34% was burned in spring. The results are consistent with the unpublished results of Hall, Loboda and Hao for average annual BC emissions of cropland fires in Russia ($0.011 \pm 0.003$ Tg yr$^{-1}$) during the period of 2003–2012. Therefore, annual BC emissions from agricultural fires in Russia are insignificant, accounting for only 1.5% of total BC emissions from fires.

## 5.       Conclusions

We have estimated daily BC emissions from forest, grassland, shrubland, and savanna fires in different geographic regions over Northern Eurasia at a 500 m $\times$500 m resolution from 2002– 2015. The results are essential for modeling the impact of BC on accelerated ice and snow melting in the Arctic. During the 14-year period, BC emissions from biomass burning in Northern Eurasia accounted for about 9.2%–9.5% of the global BC sources and 26% of the biomass burning source worldwide. Forest fires dominated BC emissions (65%) followed by grassland fires (18%). Russia was the dominant country contributing about 80% of total BC emissions from biomass burning in Northern Eurasia. Approximately 58% of the BC emissions occurred in spring time, when the greatest impact occurs on ice and snow melting in the Arctic. Our estimates of BC emissions from biomass burning were about 3.2 times higher than the GFED4.1 estimates. We attribute these differences in the mapped burned areas, fuel loading/consumption, and the emission factors. Additional atmospheric measurements of BC in regions where fires contribute the most BC emissions coupled with the modeling of atmospheric transport and deposition should help in determining which inventory best represents BC emissions.

*Acknowledgements*

We thank Vlady Alexeyev at the Sukachev Institut of Forest and Sergey Bartalev at the Space Research Institute of the Russian Academy of Sciences to provide the essential datasets for

mapping the fuel loading. This project was supported by the US Department of State, US Forest Service Research and Development, and NASA Terrestrial Ecology Program.

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

**Table Captions**

**Table 1.** Annual BC emissions in different land cover types over different geographic regions in Northern Eurasia from 2002–2015.

**Figure Captions**

**Fig. 1**. Spatial distribution of BC emissions in Northern Eurasia at a 500 m × 500 m resolution in 2003. The black line illustrates the Trans-Siberian Railway. The inset map is the geographic regions of Russia, East Asia, Central and Western Asia, and Europe.

**Fig. 2.** Comparisons of burned areas over Northern Eurasia from 2002–2015 mapped by FEI-NE, GFED4, and MCD45.

**Fig. 3**. Comparisons of burned areas from (a) 2003 and (b) 2013 in different geographic regions in Northern Eurasia mapped by FEI-NE, GFED4, and MCD45.

**Fig. 4.** Average fuel consumption for different land cover types in Northern Eurasia from 2002–2015.

**Fig. 5**. Daily BC emissions in Northern Eurasia at a 500 m × 500 m resolution from 2002–2015.

**Fig. 6.** Interannual variability of BC emissions for different land cover types in Northern Eurasia from 2002–2015.

**Fig. 7**. Daily BC emissions in Northern Eurasia from 2002–2015.

**Fig. 8.** Daily BC emissions in different land cover types in Northern Eurasia from 2002–2015. Note the differences in the Y-axis scales of BC emissions from fires in different land cover types.

**Fig. 9.** Comparisons of annual BC emissions from biomass burning in Northern Eurasia from 2002–2015 estimated by FEI-NE and GFED4.1.

**Fig. 10.** Comparisons of annual BC emissions from (a) forest and (b) non-forest fires in Northern Eurasia for FEI-NE and GFED4.1 from 2002–2015.

**Fig. 11**. Comparisons of annual BC emissions in different geographic regions in Northern Eurasia from (a) fires in FEI-NE Russia versus GFED4.1 BOAS, and (b) fires in FEI-NE Eastern Asia, Central and Western Asia, Europe versus GFED4.1 CEAS and EURO from 2002–2015.

2 **Table 1.** Annual BC emissions in different land cover types over different geographic regions in
3 Northern Eurasia from 2002–2015.

| Region | 2002 | 2003 | 2004 | 2005 | 2006 | 2007 | 2008 | 2009 | 2010 | 2011 | 2012 | 2013 | 2014 | 2015 | Total |
|---|---|---|---|---|---|---|---|---|---|---|---|---|---|---|---|
| | | | | | | Black Carbon Emissions (Gg yr$^{-1}$) | | | | | | | | | |
| **Forest** (Evergreen Needleleaf, Evergreen Broadleaf, Deciduous Needleleaf, Deciduous Broadleaf, Mixed) | | | | | | | | | | | | | | | |
| Russia | 376 | 1158 | 232 | 190 | 455 | 233 | 782 | 321 | 184 | 418 | 604 | 228 | 495 | 263 | 5939 |
| Eastern Asia | 12 | 61 | 64 | 28 | 20 | 23 | 31 | 50 | 9 | 27 | 22 | 8 | 21 | 17 | 392 |
| Central & Western Asia | 1 | 1 | 1 | 1 | 1 | 2 | 2 | 1 | 3 | 1 | 1 | 1 | 1 | 2 | 19 |
| Europe | 5 | 11 | 4 | 4 | 5 | 9 | 4 | 4 | 2 | 6 | 11 | 2 | 5 | 7 | 78 |
| Subtotal | 394 | 1231 | 302 | 223 | 481 | 266 | 820 | 376 | 197 | 451 | 638 | 239 | 522 | 288 | 6428 |
| **Grassland** | | | | | | | | | | | | | | | |
| Russia | 33 | 101 | 27 | 34 | 54 | 45 | 65 | 37 | 22 | 32 | 36 | 14 | 33 | 39 | 571 |
| Eastern Asia | 24 | 21 | 14 | 11 | 13 | 17 | 12 | 12 | 6 | 15 | 24 | 17 | 25 | 26 | 237 |
| Central & Western Asia | 108 | 118 | 119 | 76 | 106 | 59 | 84 | 49 | 77 | 23 | 52 | 14 | 43 | 43 | 970 |
| Europe | 0 | 1 | 0 | 0 | 0 | 1 | 0 | 0 | 0 | 1 | 1 | 0 | 0 | 0 | 6 |
| Subtotal | 166 | 241 | 160 | 121 | 173 | 122 | 161 | 98 | 105 | 71 | 113 | 46 | 101 | 108 | 1784 |
| **Shrubland** (Closed Shrubland and Open Shrubland) | | | | | | | | | | | | | | | |
| Russia | 40 | 172 | 12 | 41 | 22 | 18 | 39 | 39 | 48 | 29 | 62 | 57 | 23 | 22 | 624 |
| Eastern Asia | 4 | 2 | 1 | 3 | 5 | 5 | 5 | 6 | 2 | 2 | 10 | 2 | 3 | 2 | 50 |
| Central & Western Asia | 1 | 4 | 5 | 2 | 2 | 3 | 2 | 1 | 3 | 1 | 1 | 1 | 1 | 2 | 28 |
| Europe | 0 | 1 | 1 | 0 | 1 | 3 | 2 | 1 | 0 | 1 | 1 | 0 | 0 | 0 | 12 |
| Subtotal | 45 | 179 | 19 | 45 | 30 | 28 | 47 | 47 | 54 | 33 | 75 | 61 | 27 | 26 | 714 |
| **Savanna** (Woody Savanna and Savanna) | | | | | | | | | | | | | | | |
| Russia | 26 | 151 | 15 | 43 | 53 | 52 | 120 | 37 | 25 | 49 | 99 | 65 | 37 | 57 | 828 |
| Eastern Asia | 3 | 7 | 7 | 6 | 11 | 7 | 11 | 9 | 3 | 6 | 8 | 4 | 4 | 5 | 91 |
| Central & Western Asia | 2 | 3 | 2 | 2 | 3 | 4 | 3 | 4 | 2 | 1 | 3 | 2 | 3 | 4 | 38 |
| Europe | 1 | 3 | 1 | 1 | 2 | 8 | 2 | 3 | 1 | 3 | 6 | 2 | 2 | 2 | 37 |
| Subtotal | 32 | 164 | 25 | 52 | 69 | 71 | 136 | 54 | 31 | 59 | 116 | 73 | 46 | 67 | 994 |
| Total | 636 | 1815 | 506 | 441 | 752 | 488 | 1164 | 575 | 387 | 613 | 941 | 419 | 695 | 489 | 9921 |

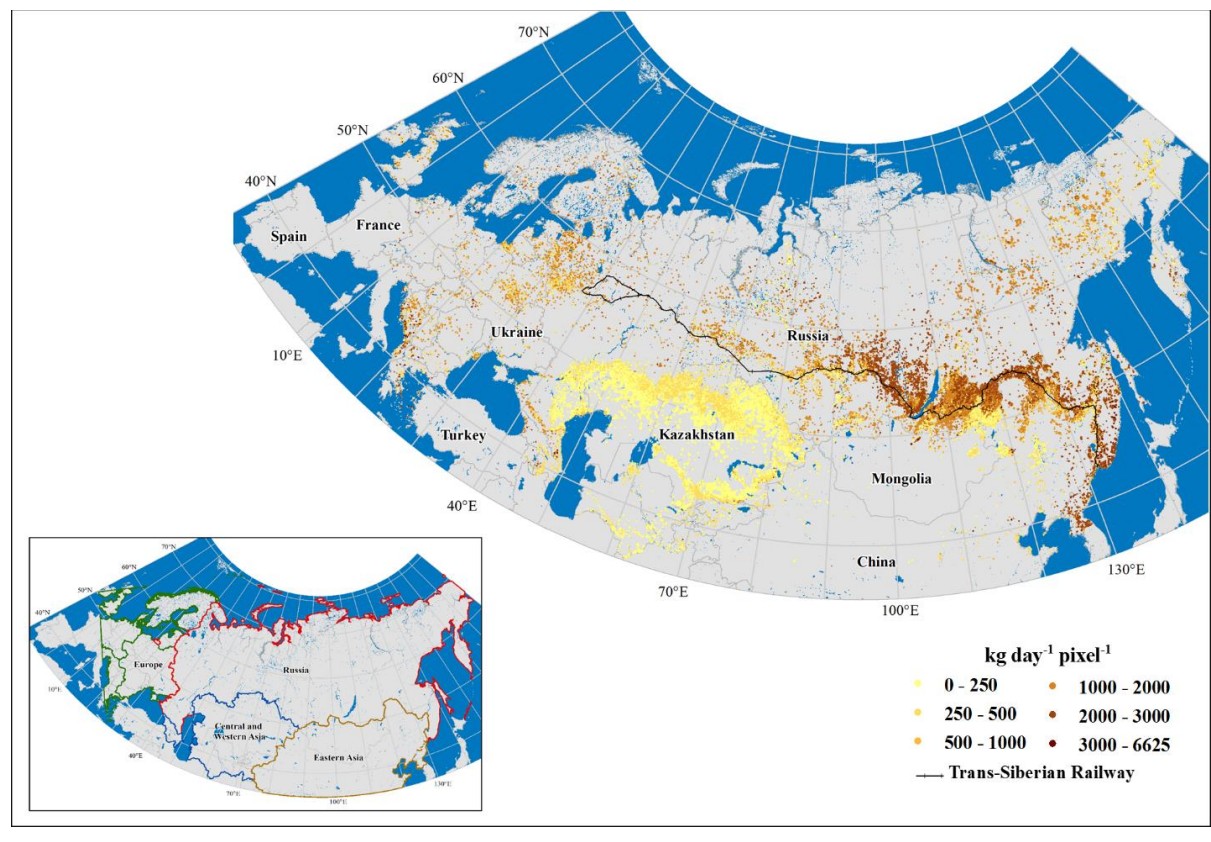

3  **Figure 1**. Spatial distribution of BC emissions in Northern Eurasia at a 500 m × 500 m
4  resolution in 2003. The black line illustrates the Trans-Siberian Railway. The inset map is the
5  geographic regions of Russia, East Asia, Central and Western Asia, and Europe.

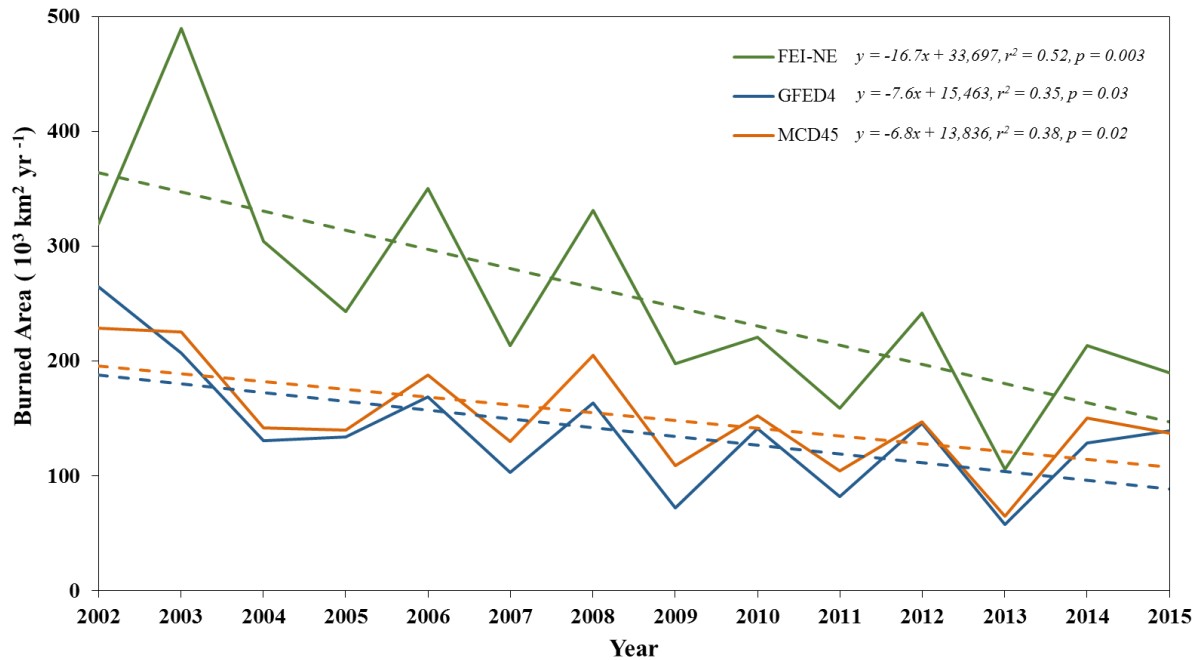

Figure 2. Comparisons of burned areas over Northern Eurasia from 2002–2015 mapped by FEI-NE, GFED4, and MCD45.

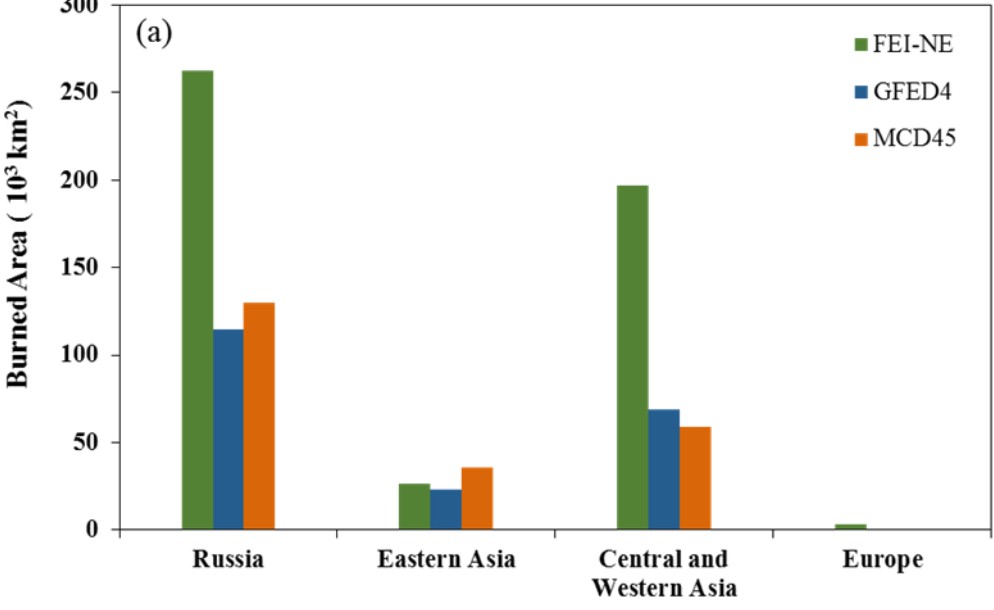

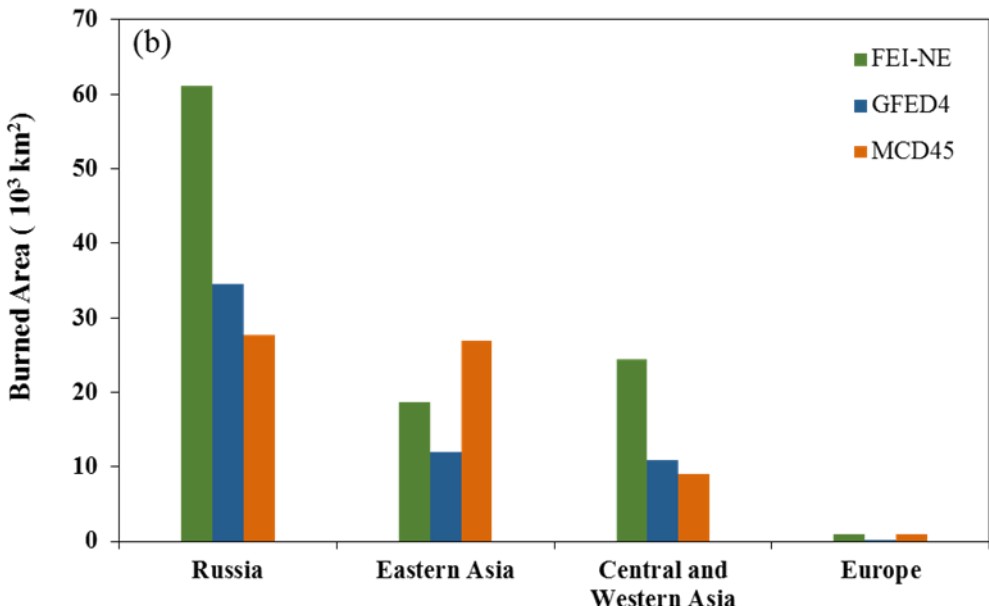

**Figure 3.** Comparisons of burned areas from (a) 2003 and (b) 2013 in different geographic regions in Northern Eurasia mapped by FEI-NE, GFED4, and MCD45.

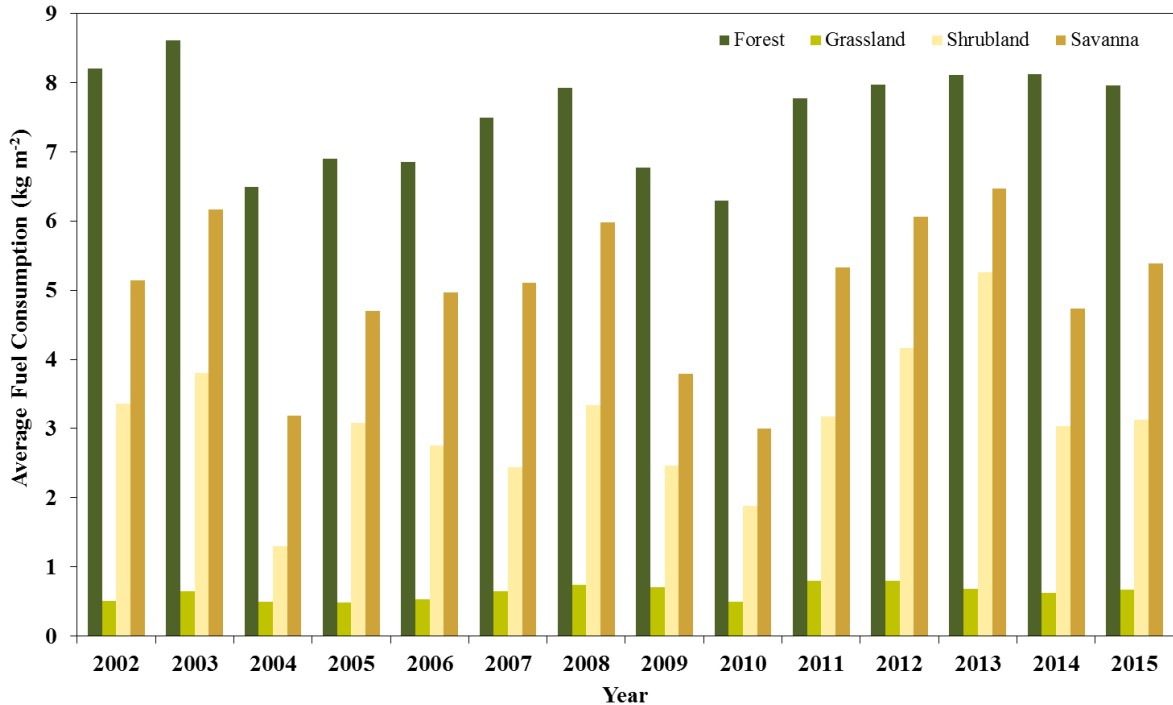

2   **Figure 4.** Average fuel consumption for different land cover types in Northern Eurasia from
3   2002–2015.

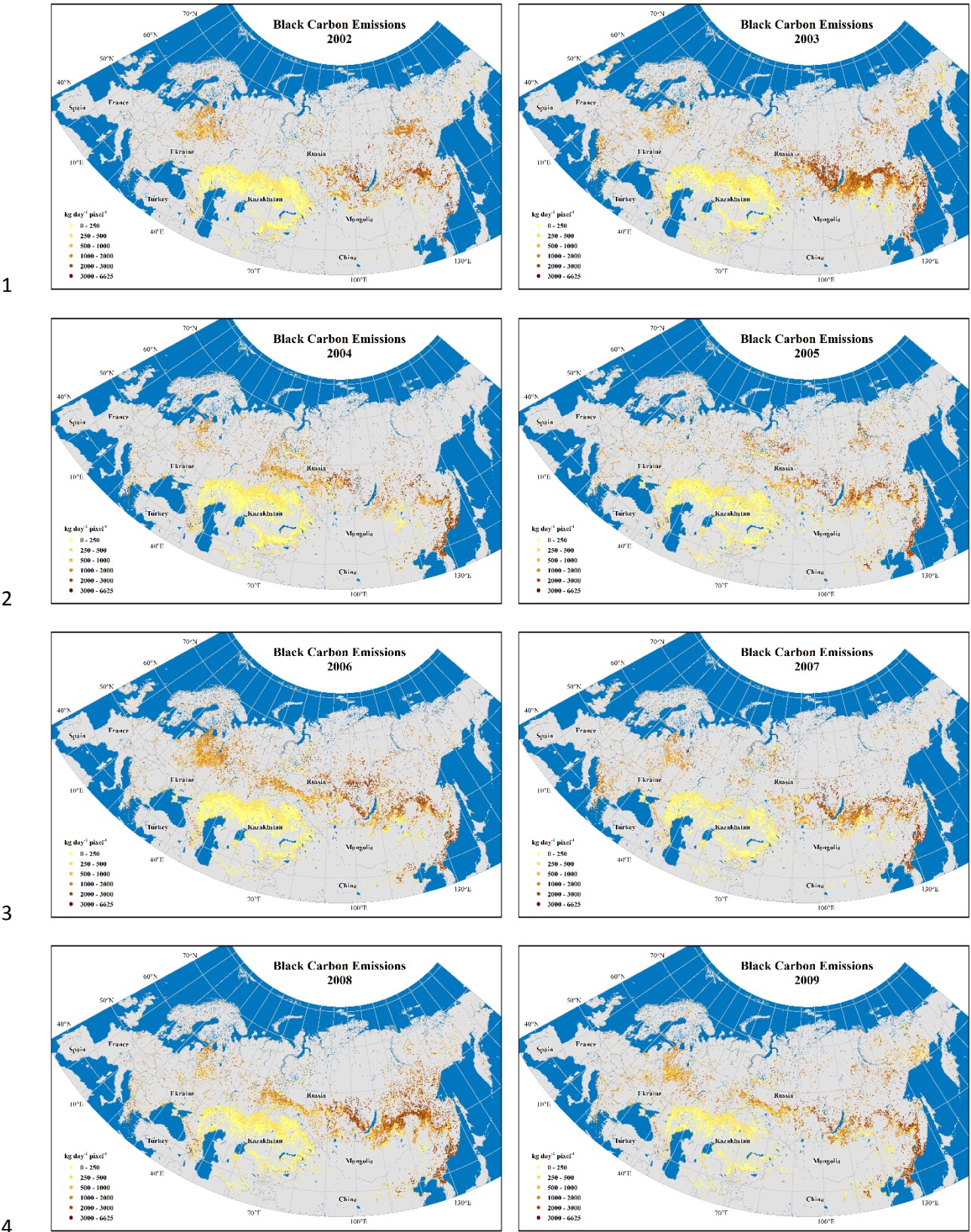

**Figure 5.** Daily BC emissions in Northern Eurasia at a 500 m × 500 m resolution from 2002–2015.

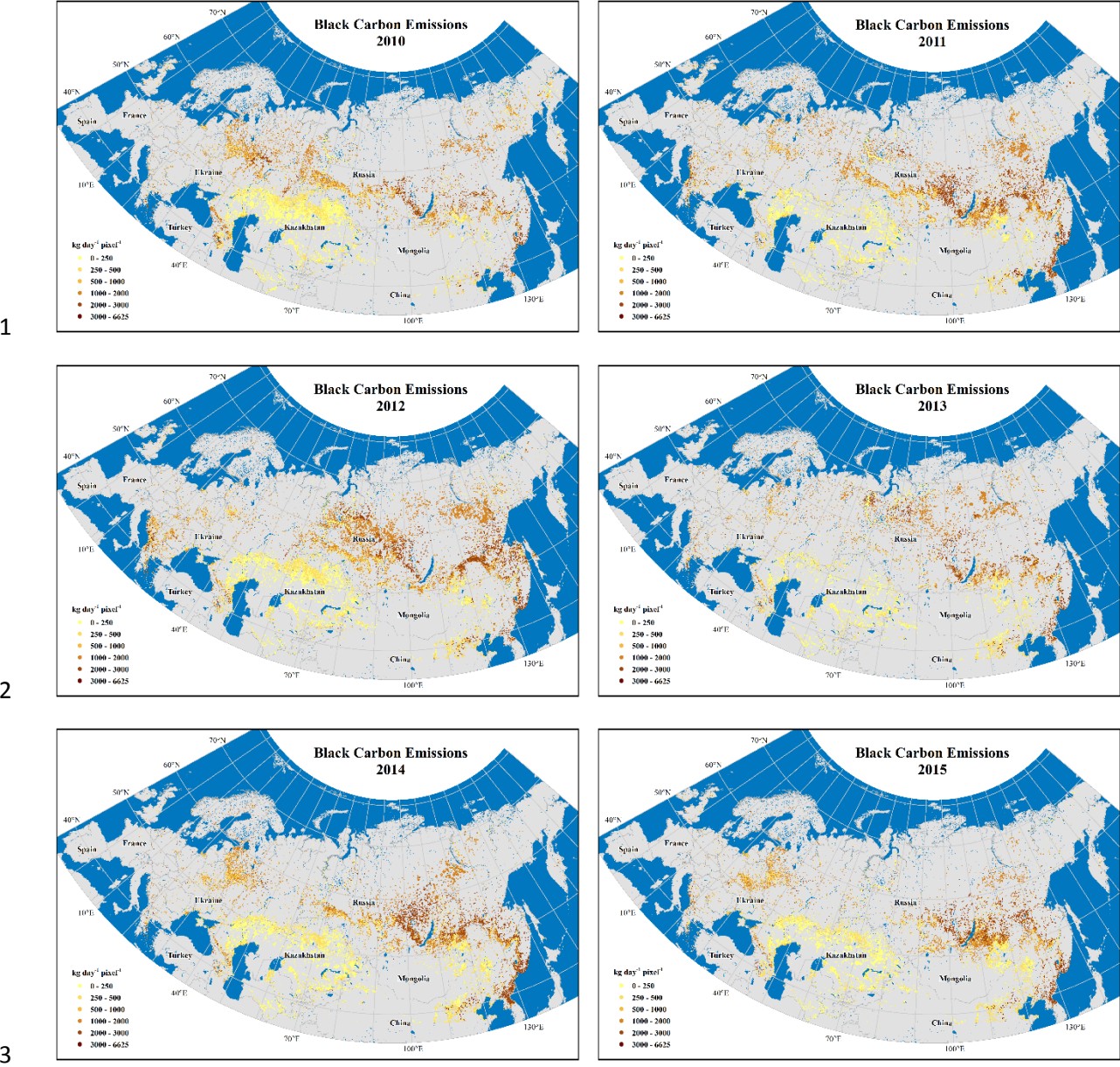

**Figure 5.** Daily BC emissions in Northern Eurasia at a 500 m × 500 m resolution from 2002–2015 (continued).

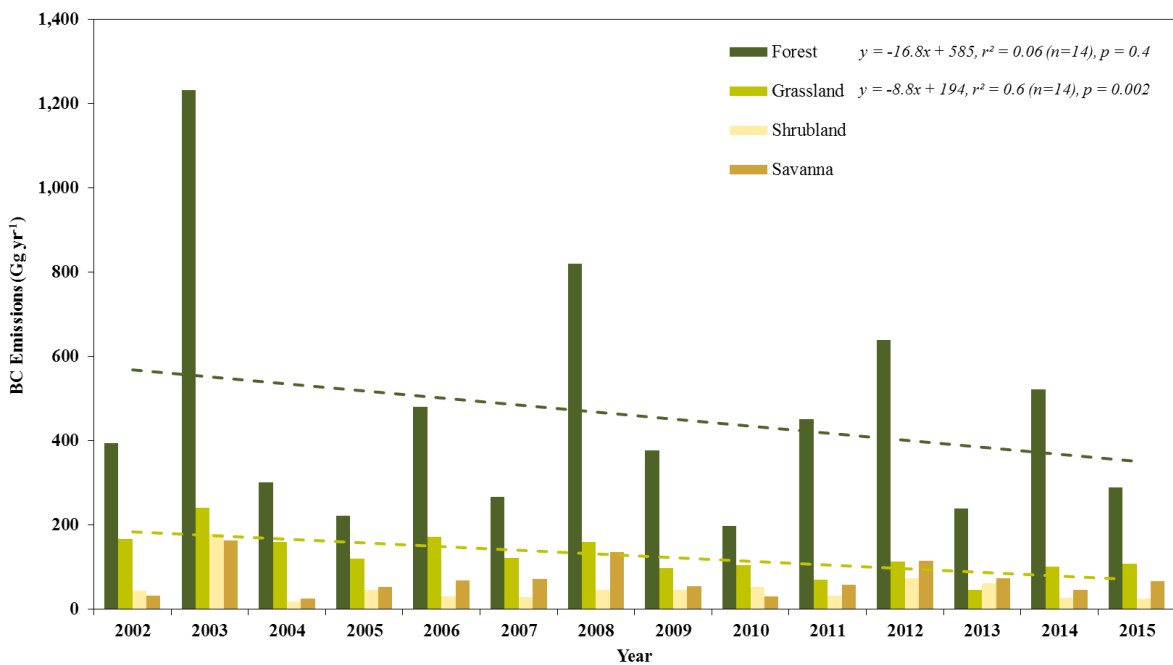

**Figure 6.** Interannual variability of BC emissions for different land cover types in Northern
Eurasia from 2002–2015.

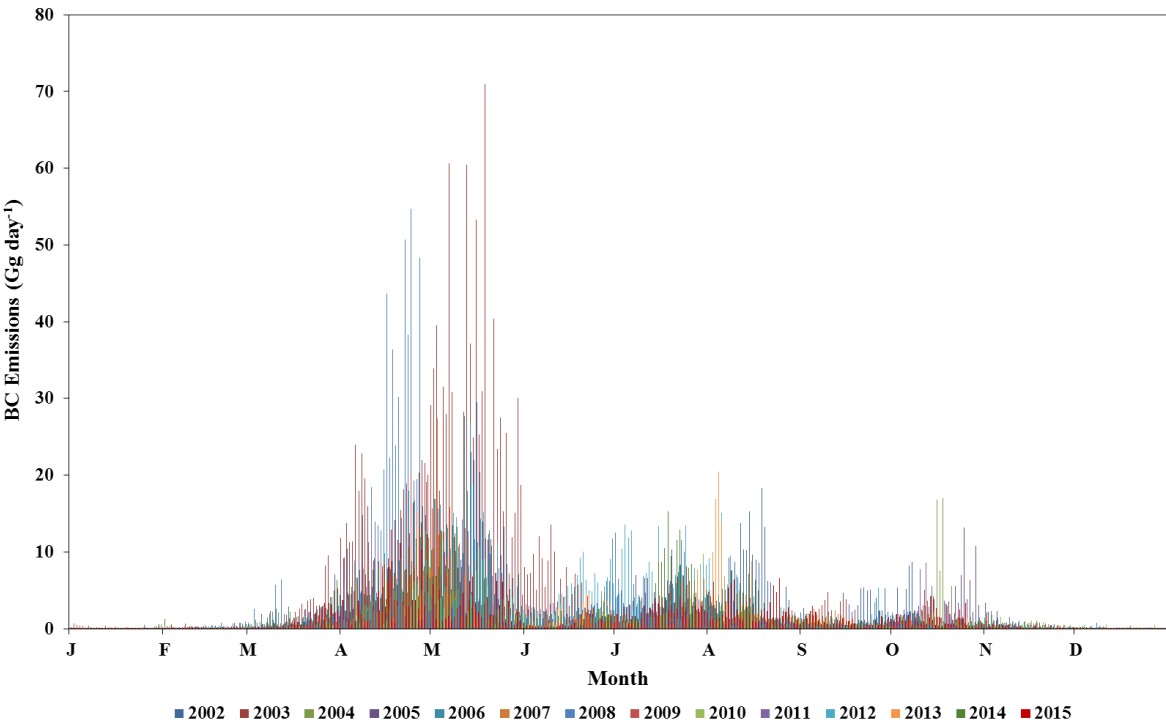

2  **Figure 7.** Daily BC emissions in Northern Eurasia from 2002–2015.

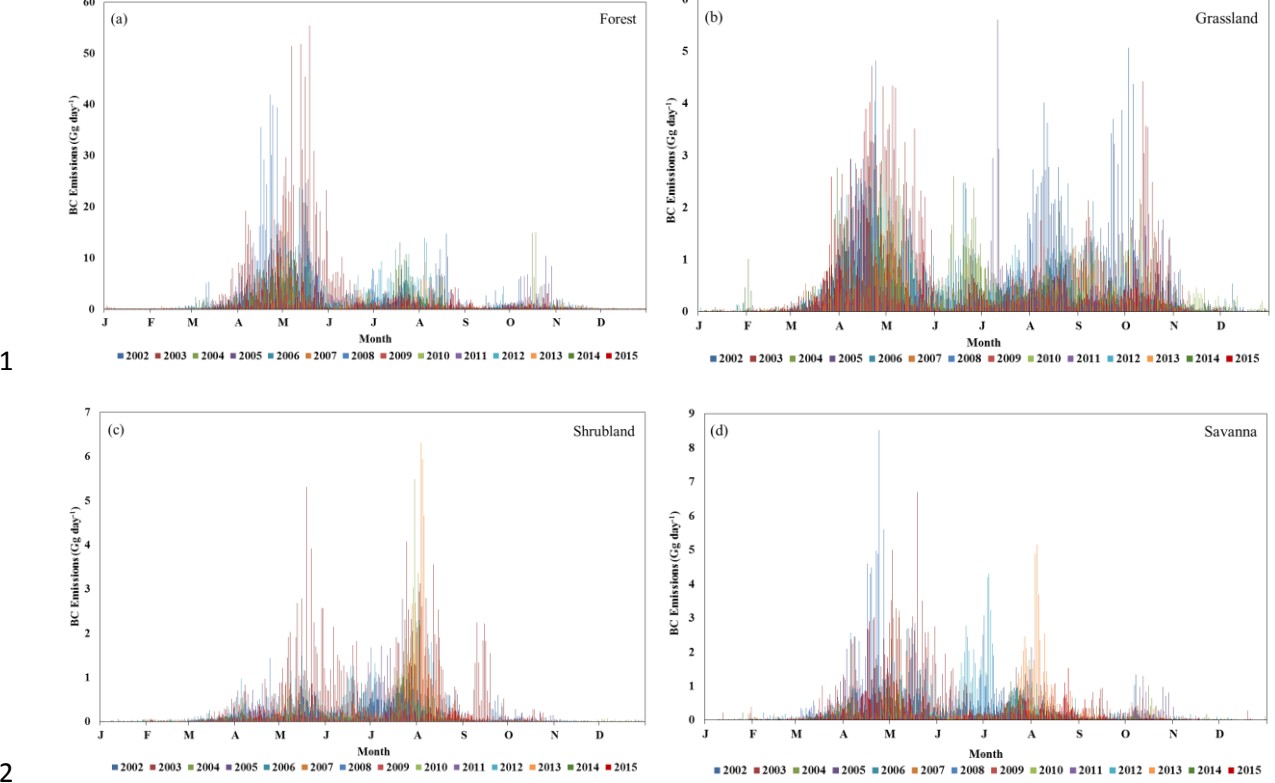

**Figure 8.** Daily BC emissions in different land cover types in Northern Eurasia from 2002–2015.
Note the differences in the Y-axis scales of BC emissions from fires in different land cover
types.

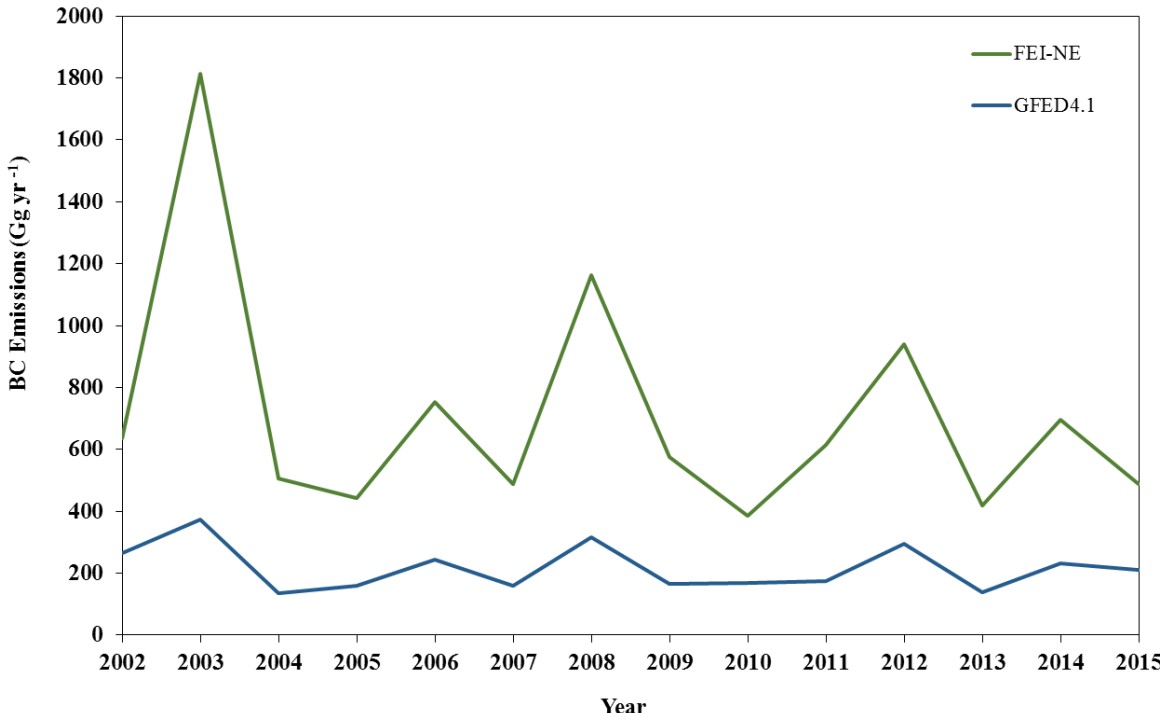

2  **Figure 9.** Comparisons of annual BC emissions from biomass burning in Northern Eurasia from
3  2002–2015 estimated by FEI-NE and GFED4.1.

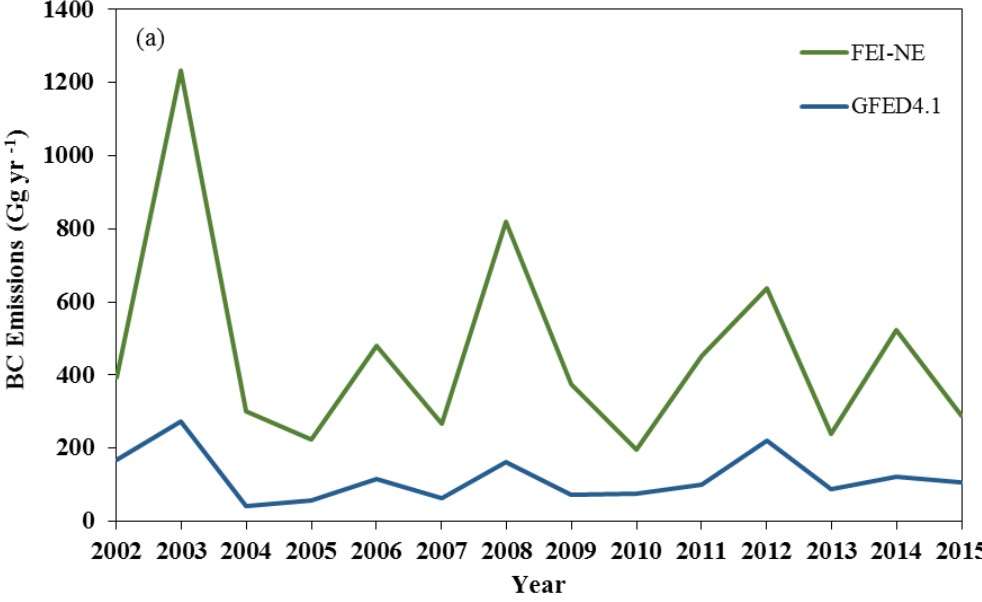

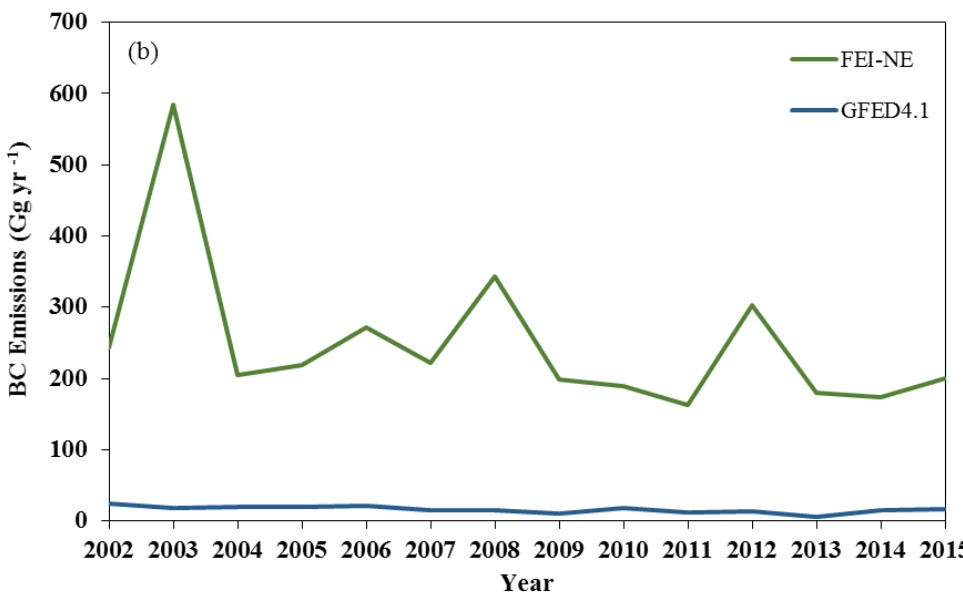

**Figure 10.** Comparisons of annual BC emissions from (a) forest and (b) non-forest fires in
Northern Eurasia for FEI-NE and GFED4.1 from 2002–2015.

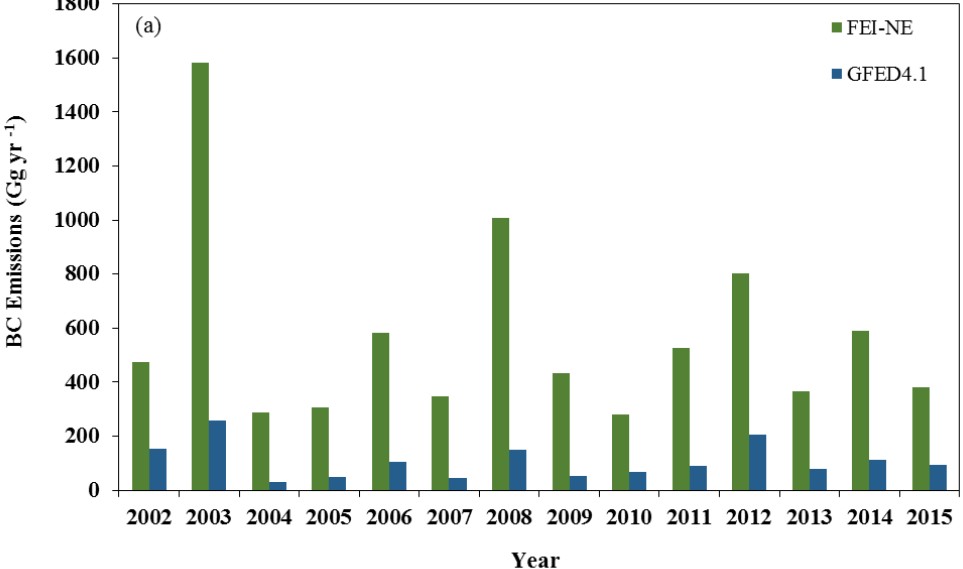

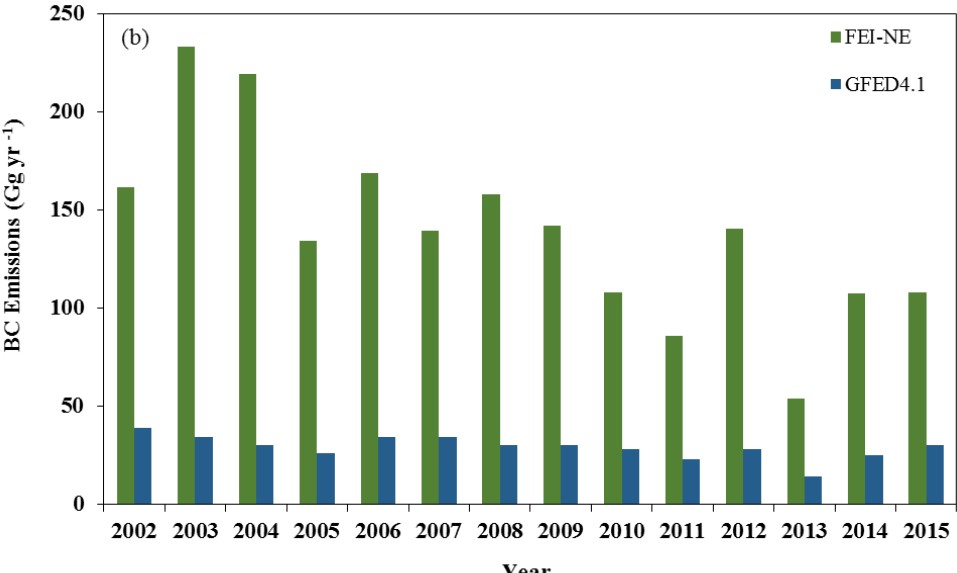

**Figure 11.** Comparisons of annual BC emissions in different geographic regions in Northern Eurasia from (a) fires in FEI-NE Russia versus GFED4.1 BOAS, and (b) fires in FEI-NE Eastern Asia, Central and Western Asia, Europe versus GFED4.1 CEAS and EURO from 2002–2015.