# Peer review of "Daily black carbon emissions from fires in Northern Eurasia for 2002–2015"

_Geoscientific Model Development, 2016_

## Short Comment (SC1) · 22 Apr 2016

Dear authors,

In my role as Executive editor of GMD, I would like to bring to your attention our Editorial version 1.1:

http://www.geosci-model-dev.net/8/3487/2015/gmd-8-3487-2015.html

This highlights some requirements of papers published in GMD, which is also available on the GMD website in the 'Manuscript Types' section:

http://www.geoscientific-model-development.net/submission/manuscript_types.html

In particular, please note that for your paper, the following requirements have not been met in the Discussions paper:

[Figure]

- "Inclusion of Code and/or data availability sections is mandatory for all papers and should be located at the end of the article, after the conclusions, and before any appendices or acknowledgments. For more details refer to the code and data policy" (Editorial v1.1, Appendix A1)

- "Papers describing data sets designed for the support and evaluation of model simulations are within scope. These data sets may be syntheses of data which have been published elsewhere. The data sets must also be made available, and any code used to create the syntheses should also be made available." (Editorial v1.1, Appendix A5)
  For these papers the same criteria as for model description papers apply, i.e., "The main paper must give the model name and version number (or other unique identifier) in the title." (Editorial v1.1, Appendix A2) In this case the "model" is the "data set".

Please add a data availability section and include the data sets name and version number in the title in your revised submission to GMD.

Yours,

Astrid Kerkweg

---

## Author Comment (AC1) · 25 Apr 2016

Reply: The data set will be made available through the U.S. Forest Service Research Data Archive: http://www.fs.usda.gov/rds/archive/

The US Forest Service Research Data Archive contains the data set, complete metadata for the data set(s), and any other documentation the researcher deemed important to understanding the data set(s). Each data is assigned a digital object identifier (DOI) identifying the content and providing a persistent link to its location on the Internet. The process of submitting / publishing a data set on the US Forest Service Research Data Archive typically requires two to three months. We have initiated the process.

Questions: 1. Do we need to submit a data set temporarily available during the review process?

[Figure]

2. Must our data set be available in the US Forest Research Data Archive prior to publication in GMD?

I look forward to your advice.

Thanks, Wei Min Hao

---

## Short Comment (SC2) · 26 Apr 2016

Thanks for making the data available in a public archive and citable via DOI!

*1. Do we need to submit a data set temporarily available during the review process?*
This would, of course, improve the review process. At least the editor and the referees should get access to the data, if they want to. Anyhow, making the data available to the public is not yet a vital necessity for GMD articles, however very much appreciated.

*2. Must our data set be available in the US Forest Research Data Archive prior to publication in GMD?* Again, it would be good to have it on-line at the point of final publication. Especially, the DOIs should be cited in the article. (I do not know, if you know the DOIs prior to the actual publication of the data sets). Your estimate for the

data to be on-line was 2-3 month. At the end of the discussion phase nearly 2 month of this time are up, adding some extra time for your revision of the article, the editor decision and the type-setting of the manuscript, the data sets should be online at the time of final publication or at least no long delay should be caused by waiting for the data to go online.

Best, Astrid Kerkweg

---

## Referee Comment (RC1) · Anonymous Referee #1 · 15 May 2016

Comments to "Daily black carbon emissions from fires in Northern Eurasia from 2002 to 2013 " by Hao et al.

Hao et al. presented a new data set of black carbon emissions from open biomass burning for northern Eurasia based on a burned area algorithm using MODIS data as input. The spatial (500m) and temporal (daily) resolutions of this data set are high enough to make it useful for regional and global atmospheric studies thus it represents a valuable contribution. They also examined the seasonal patterns for fire activity and BC emissions for different land cove types and shed insights on the interactions of forest fire, spring snow dynamics and arctic ice dynamics. I find generally the manuscript well prepared and easy to follow with all methods and data sets used being clearly described.

[Figure]

Where possible, the derived burned area and BC emissions are compared with other data sets (such as MCD45 and GFED4). The authors found significantly higher burned area and BC emissions than GFED4, which is now widely used in the fire and atmospheric studies. The central objective of this study is to present the new BC emissions data set derived using a bottom-up approach. It's important that validation information of burned area, comparison with other data sets regarding derived biomass consumption in fires, and uncertainty information of BC emissions should be sufficiently discussed. These points are detailed in the general comments as below. I recommend the manuscript being accepted after the authors adequately address these points.

General comments:

- The burned areas derived by the authors are in close agreement with MCD45 data, this is very good. I believe more details regarding validation of the algorithms for burned area used in the study could be useful, for example the error characterization process and the error information (commission and omission errors) obtained when validating burned area with high-resolution reference BA data. So far only one reference is (Hao et al. 2014) however this one is not found in the reference list.

- In section 4.1 the BC emissions are compared with GFED data. The BC emissions derived in this study much higher than GFED4. However here mainly simple comparisons are presented. The BA from all land cover types in this study is roughly twice that of GFED4, however total BC emissions are 3.5 times higher if I understand well and so the ratio is much higher than BA. Is this because of the higher emission factors used or higher fuel consumption rates? It would be nice to present the typical fuel consumption rates in fire for different land cover types the authors have obtained before converting to BC emissions. Then the readers might have better understanding how these differences arise.

- I understand it could be difficult to generate completely quantitative uncertainty information for BC emissions using error propagation because statistical distribution as-

sumptions then have to be made regarding major input variables in the equation to calculate BC emission. However a general discussion on uncertainty of derived BC emissions is still useful to guide potential data users.

Technical comments:

P3-line 27: I don't find the Hao et al., 2014 in the reference list, this is a very important reference as burned area algorithm should be validated there.

Figure 2 – "forests and non-forests" are both explicitly included in the caption of this figure. I find this a little misleading because one may expect that burned area for forests is shown separately with that for non-forests from reading the title but this is note the case.

P6-line16-17: what is the trend for BC emissions over forest?

P6-line27-29: are these grassland fire emissions with bimodal temporal distributions are also spatially separated?

P7-line 9: The "Evangeliou et al., 10 this issue " is missing in the reference list.

P7-line16-17: I don't fully understand here why BC emissions from agricultural fires are excluded when comparing GFED4 and GFED3. Comparison of BC emissions between GFED4 and GFED3 excluding agricultural fires is a little distracting here because all remaining parts in the same paragraph focus on the total emissions from all land cover types.

---

## Author Comment (AC2) · 17 May 2016

The comments are very useful. They will be incorporated in the revised manuscript. Thanks!

---

## Referee Comment (RC2) · Anonymous Referee #2 · 26 May 2016

This manuscript develops a new data set of BC emissions from fires over Northern Eurasia from 2002 to 2013, which is useful for the atmospheric modeling of BC. I recommend that this paper can be accepted for publication after making data available for the community. Please also see some comments below. Comments: 1) Line 1-3/Page 2: Please cite the latest IPCC report for the role of BC among all climate forcers, rather than only one literature studying BC only. Black carbon should not be the second most important species for climate forcing after CO2 in our latest understanding. 2) Line 7-9: Please review the literature estimating the emissions of BC, rather than citing just one literature on this. There are many recent important studies developing global emission inventories of BC, which are not noticed by the authors. Please improve this part. In addition, "an average of 7.5 Tg yr-1" is not clear. Please clarify it. 3) Introduction: Please provide a summary of previous studies estimating the black carbon emissions

from fires and summarize the differences with the present work. 4) Figure 2. This figure does not well illustrate the differences between FEI-NE and the other products cited. Please show the geographic distributions of the differences of burned areas between FEI-NE and GFED4 and between FEI-NE and MCD45. 5) Figure 7: This figure does not well illustrate the differences between FEI-NE and the other products cited. Please show the geographic distributions of the differences of BC emissions between FEI-NE and GFED3 and between FEI-NE and GFED4.

---

## Author Comment (AC3) · 26 May 2016

All the comments of reviewer 2 will be incorporated in the revision. Thanks.

---

## Author Comment (AC4) · 21 Jul 2016

I request to extend the submission of the revised manuscript to September 15, 2016. The delay is caused by reprocessing and archiving the dataset which takes longer than expected. Please let me know your decision. Thanks a lot. Wei Min Hao

---

## Author Comment (AC5) · 14 Sep 2016

I request to delay the resubmission of the manuscript to October 31, 2016. We are in the process of extending the years studied of 2002-2013 to 2002-2015. it also takes longer to archive the dataset, which depends on the schedule of the Forest Service Archive Office.

Please let know your decision soon. Thanks a lot. Wei Min Hao

———————————

---

## Author Response (AR1)

The manuscript has expanded to include two additional years (2014 and 2015) with the most updated dataset. The new title is "Daily black carbon emissions from fires in Northern Eurasia for 2002–2015." The revised text is also reflected to the analysis and discussion of the 14-year dataset instead of the original 12-year dataset. The archived dataset will be available on a web site shortly.

The specific responses to the referee's comments are summarized below:

**Referee #1 Comments and Responses**

General comments

- Validation of burned areas (BA):
  Response: It is a typo of Hao et al. 2014 and should be Hao et al. 2012 which is included in the reference list. A description of the original validation of the algorithm in western U.S. has been described in detail by Urbanski et al. (2011) (p. 3, line 25-36), and the correlation of FEI-NE BA with Landsat images is summarized by Hao et al. (2012) (p. 4, line 1-8).

- Fuel consumption and emission factors:
  Response: Our FEI-NE burned areas were 1.8 times higher than those of GFED4 (p. 6, line 12-14), but the FEI-NE black carbon (BC) emissions from fires were 3.2 times higher than the GFED4.1 BC emissions (p. 8, line 27). The differences in fuel consumption (p. 6, section 3.2, line 24-31, and Fig. 4) and emission factors (p. 8, section 4.1, line 35 & 36 – p. 9, line 1-2) of the two datasets are the other factors contributing to the discrepancy of the BC emissions from fires. It is difficult to quantify the exact differences for the two datasets because each one is integrated with different land cover types from numerous 500m x 500m or 0.25° x 0.25° grid cells over the entire Northern Eurasia. The geographic regions covered by FEI-NE and GFED4 are also close but are not exactly the same.

- Uncertainty:
  Response: The section 2.6 (p. 5, line 25-32) is added to discuss the estimation of uncertainty.

Technical comments

- P 3-line 27: Hao et al., 2014
  Response: It is a typo of Hao et al. 2014 and should be Hao et al. 2012, which is included in the reference list (p. 12, line 25-27 – p. 13, line 1-2).

- Figure 2: Forests and non-forests
  Response: The figure caption is revised to "Comparisons of burned areas over Northern Eurasia from 2002–2015 mapped by FEI-NE, GFED4, and MCD45," so as to avoid the confusion of the term "forests and non-forests." (p. 20)

- P 6-line 16-17: trend for BC emissions over forest

Response: There is no apparent trend of BC emissions over forests (p. 7, line 21-23)

- P 6-line 27-29: grassland fire emissions
  Response: It is for the entire Northern Eurasia, so "Northern Eurasia" is added to the text to make it clear (p. 8, line 3; p. 9, line 26).

- P 7-line 9: Evangeliou
  Response: The reference of "Evangeliou et al., 2016" has been updated in the reference list (p. 12, line 5-9).

- P 7-line 16-17: agricultural fires
  Response: The comparison of FEI-NE with GFED3 is removed, because the GFED3 dataset is no longer available on the GFED web site. The words "excluding agricultural fires" is added to the beginning of the paragraph of "3 Results" (p. 5, line 34-35) for all the following texts. "excluding agricultural fires" is also added again to make it more clear (p. 8, line 23-24).

**Referee #2 Comments and Responses**

1) Line 1-3/Page 2: references on the roles of BC as climate forcers
   Response: three additional references are included: IPCC, 2013; Stohl et al., 2015; Sand et al., 2016. (p. 2, line 3). Also, change BC is the "second most" species to "one of the leading" species (p. 2, line 2).

2) Line 7-9/Page 2: literature review
   Response: Extensive reviews of the most recent BC emission inventory are made: Bond et al., 2013 and Wang et al., 2014 (p. 2, line 6-8). An earlier review of Bond et al. (2004) is also added (p. 2, line 8-9). An average of 7.5 Tg yr$^{-1}$ is also made clear. All the revisions are summarized in the first paragraph of p. 2, line 2-13.

3) Introduction: previous and present work
   Response: The revisions are in the first paragraph (p. 2, line 2-13), the second paragraph (p. 2, line 23-33), and the last paragraph (p. 2, line 34-39) of page 2, and the first paragraph of page 3 (p. 3, line 1-7).

4) Figure 2 geographic differences of burned areas between FEI-NE, GFED4, and MCD45
   Response: Fig. 2 is re-drawn. The geographic differences of burned areas for FEI-NE, GFED4, and MCD45 in 2003 (the largest difference) and in 2013 (the smallest difference) for Russia, Eastern Asia, Central and Western Asia, and Europe are shown in Figs. 3a and 3b and discussed in section 3.1 (p. 6, line 15-22).

5) Figure 7: the differences in the geographic distributions of BC emissions between FEI-NE, GFED3, and GFED4
   Response: GFED3 is not compared because the dataset is no longer available on the GFED web site. The differences in the geographic distributions of FEI-NE and GFED4 are illustrated in Figs. 11a and 11b.and were discussed in the last paragraph of section 4.1 (p. 9, line 3-15).

[revised manuscript text omitted]